# Rapid systematic review on developing web-based interventions to support people affected by cancer

Samuel Cooke  ,[1] David Nelson,[2,3] Heidi Green,[1] Kathie McPeake,[3,4] Mark Gussy,[2] Ros Kane[1]

¹School of Health and Social Care, University of Lincoln, Lincoln, UK
²Lincoln International Institute for Rural Health, University of Lincoln, Lincoln, UK
³Macmillan Cancer Support, London, UK
⁴NHS Lincolnshire Clinical Commissioning Group, Lincoln, UK

**Correspondence to**
Dr Samuel Cooke;
scooke@lincoln.ac.uk

## ABSTRACT

**Objective** To systematically identify and explore the existing evidence to inform the development of web-based interventions to support people affected by cancer (PABC).

**Design** A rapid review design was employed in accordance with the guidance produced by the Cochrane Rapid Reviews Methods Group and reported using the Preferred Reporting Items for Systematic Reviews and Meta-Analyses checklist. A rapid review was chosen due to the need for a timely evidence synthesis to underpin the subsequent development of a digital resource (Shared Lives: Cancer) as part of an ongoing funded project.

**Methods and outcomes** Keyword searches were performed in MEDLINE to identify peer-reviewed literature that reported primary data on the development of web-based interventions designed to support PABC. The review included peer-reviewed studies published in English with no limits set on publication date or geography. Key outcomes included any primary data that reported on the design, usability, feasibility, acceptability, functionality and user experience of web-based resource development.

**Results** Ten studies were identified that met the pre-specified eligibility criteria. All studies employed an iterative, co-design approach underpinned by either quantitative, qualitative or mixed methods. The findings were grouped into the following overarching themes: (1) exploring current evidence, guidelines and theory, (2) identifying user needs and preferences and (3) evaluating the usability, feasibility and acceptability of resources. Resources should be informed by the experiences of a wide range of end-users taking into consideration current guidelines and theory early in the design process. Resource design and content should be developed around the user's needs and preferences and evaluated through usability, feasibility or acceptability testing using quantitative, qualitative or mixed methods.

**Conclusion** The findings of this rapid review provide novel methodological insights into the approaches used to design web-based interventions to support PABC. Our findings have the potential to inform and guide researchers when considering the development of future digital health resources.

**Trial registration number** The review protocol was registered on the Open Science Framework (https://osf.io/ucvsz).

## STRENGTHS AND LIMITATIONS OF THIS STUDY

⇒ This review provides a rapid, yet comprehensive synthesis of the current evidence to support the time-sensitive decision making for the development and implementation of a novel digital resource (Shared Lives: Cancer) to help support people affected by cancer.

⇒ This rapid review, while streamlined, was conducted using a systematic methodology, following rigorous reporting guidelines to ensure transparency and reproducibility.

⇒ While considered a key part of the knowledge synthesis 'family', rapid review methods are not subject to the same robustness as a full systematic review and are more vulnerable to bias and error.

⇒ Due to time constraints, database searches were restricted to one database only and no formal quality assessment was performed on the included studies.

## INTRODUCTION

Improvements in cancer screening, early detection, diagnostic methods and treatment are resulting in an increasing number of people living with and beyond cancer.[1–4] Globally, there were an estimated 18.1 million new diagnoses in 2018.[5] In the UK, it is estimated that 4 million people will be living with and beyond cancer by 2030.[6] As services have expanded to support the continuing rise in cancer incidence, so too have the complexities in delivering care.[7–9] This is epitomised by the changes in the way cancer care has been implemented over recent decades, which in the UK, for example, now involves a multitude of bodies responsible for purchasing, commissioning, delivering and regulating services.[8 10]

To ensure the provision of future cancer services adapts to changes in health needs, medical advances and societal developments, the national health service (NHS) England implemented a long-term plan in which digital health technologies are central.[11] Digital health technologies have become an

important tool in cancer care with the potential to revolutionise patient data, transform patient experiences, improve patient recovery and improve the access, integration and personalisation of care.[9–12] Evidence suggests that individuals living with and beyond cancer are engaging with digital health technologies now more than ever[13–16] and are using them to frequently access online health information as well as virtual support groups and forums.[17 18]

The rapid growth of internet use has led to a substantial increase in the number of web-based interventions to support people affected by cancer (PABC), including a wide range of educational and psychosocial platforms,[19–21] social media sites,[22] mobile applications[22 23] and digital health interventions that focus on specific health behaviours, for example, physical activity and diet.[24] While previous reviews have focused predominantly on the evaluation of web-based interventions, there remains little evidence documenting the developmental (design, usability, feasibility, acceptability, functionality and user experience) processes of web-based interventions in cancer populations. This review assumes a novel approach by exploring and synthesising the academic literature that reports on the development of web-based resources that support PABC. This will explicitly include resources designed to support the physical, mental and social consequences of cancer.

The findings will be used to directly inform the development of a novel web-based resource called (Shared Lives: Cancer),[25 26] that aims to support PABC through making qualitative research data on lived cancer experience publicly available and freely accessible.

This rapid review aims to:

► Identify and map the peer-reviewed academic evidence that reports primary data concerning the development of web-based interventions for supporting PABC.
► Collate and report on the academic evidence with a view to informing web-based interventions for supporting PABC.

## METHODS

This study used a rapid review approach adhering to the recently published guidance from the Cochrane Rapid Reviews Methods Group and for reporting used the Preferred Reporting Item for Systematic Reviews and Meta-Analyses (PRISMA) checklist, see online supplemental material S1. Rapid reviews are now considered a key component of the knowledge synthesis family alongside systematic reviews, scoping reviews and realist reviews. They provide a streamlined, efficient and pragmatic approach to evidence synthesis.[27] In summary, rapid reviews are a form of evidence synthesis in which components of the systematic review process are simplified, with a view to producing findings in a timely manner.[28 29] Still, rapid reviews must remain systematic in their approach and have a duty to report their methods in a transparent manner making sure they are clear about deviations or omissions from the PRISMA criteria. This review was

conducted over a 4-month period (July 2021–October 2021). The study protocol has been registered on the Open Science Framework (osf.io/ucvsz) to promote reproducibility and facilitate methodological transparency, see online supplemental material S2.

### Patient and public involvement
No patient involvement.

### Search strategy
Keyword searches together with Boolean operators (OR and AND) and truncation (*) were used to locate relevant peer-reviewed literature on the development of web-based support that is delivered to PABC. Due to the need to produce findings in a timely manner, database searches were limited to one database which is considered acceptable for a rapid systematic review. MEDLINE was searched as it is the leading full-text database of biomedical and health journals. The primary search strategy and syntax were developed and refined by three members of the review team (SC, DN, HG). All database searches were supplemented by Google Scholar searches in addition to forward and backward citation tracking on all relevant articles. Database searches were continually updated to identify and incorporate the most up to date evidence where appropriate.

To identify PABC, the following keywords were used: "cancer surviv*" or "living with cancer" or "living with and beyond cancer" or "cancer patient*" or "patients with cancer" or "people affected by cancer" or "oncology patient" or "cancer experience*" or "cancer management" or "cancer support" or "cancer care*". To identify web-based support and interventions, the following keywords were used: "web*" or "internet*" or "online*" or "digital*". To search literature on user experience, the following keywords were used: "user experience*" or "usability" or "functionality" or "design" or "interaction" or "development" or "user testing". The search strategy for MEDLINE can be found in online supplemental material S2.

All retrieved records were collated and stored using Endnote referencing software (EndNote V.X9, Clarivate Analytics, Philadelphia, Pennsylvania, USA). The titles and abstracts were screened against the eligibility criteria by one reviewer (SC). Where there was uncertainty about the inclusion of an article after title and abstract screening, the first author (SC) discussed this with the second author (DN) to reach a final decision. Following title and abstract screening, the remaining articles were independently screened by full text, for inclusion by two reviewers (SC and DN), with any disagreements again resolved through discussion.

### Eligibility criteria
#### Inclusion criteria
Peer-reviewed publications were selected for inclusion in this review if they met the following pre-defined eligibility based on the PICOT approach. Population: adults (aged

18+), all genders, people living with cancer or affected by cancer, caregivers, any geographical location. Intervention: website-based cancer support resources. Comparator: not applicable. Outcomes: reports primary data on the design, usability, feasibility, acceptability, functionality or user and developer experience of web-based support for PABC. Type: reports empirical research data using either quantitative, qualitative or mixed methods design. Only publications written in English language were included.

### Exclusion criteria

Peer-reviewed publications were excluded based on the following exclusion criteria. Population: non-adult population (under the age of 18). Intervention: support programmes that focus solely on mobile and digital apps, E-learning programmes or interventions (self-directed and practitioner/professionally led), social media or networking sites. Comparator: not applicable. Outcomes: no primary data reported on the design, usability, feasibility, acceptability, functionality or user experiences of web-based support for PABC. Type: systematic reviews or literature reviews, editorials, commentaries, opinion pieces, case series or reports.

### Data abstraction

Data were extracted using an adapted Cochrane Data Extraction Template, see online supplemental material S3. One reviewer (SC) undertook data extraction for each full text article with cross checking taking place by a second reviewer (DN). Study characteristics were extracted from each study based on (1) study methods (eg, aims/objectives, study design, participants, outcomes), (2) details on the web-based intervention/support and (3) study findings (details of all relevant data concerning user experience, needs, preferences, usability, acceptability, feasibility, functionality and design).

### Quality assessment

The focus of this rapid review is on identifying and exploring the literature on the development of web-based support that is delivered to PABC, therefore, a quality assessment of included articles was not deemed appropriate. The omission of a quality assessment was in line with the methodological approach taken by other rapid systematic reviews where the focus is on producing evidence quickly.[30]

### Data synthesis and analysis

The review included a wide range of study designs that used quantitative, qualitative and mixed methodologies. To identify and map the evidence on the development of web-based interventions for supporting PABC, we tabulated the results. This was then accompanied by a narrative summary where comments on the similarities and dissimilarities within data were made. Due to the wide heterogeneity of the design and outcomes of included studies, as well as the considerable amount of qualitative data, a formal statistical meta-analysis was not conducted; however, the findings were synthesised narratively.

## RESULTS

### Search results

The search of MEDLINE database provided a total of 2446 distinct citations with an additional 6 identified through secondary sources, see figure 1. After reviewing for title and abstract, 2439 did not meet the pre-specified eligibility criteria. The remaining 13 citations were reviewed for full text and examined in detail for inclusion in this review. Three did not meet the pre-specified inclusion criteria as these were self-help, psychological and educational supportive interventions. The resource the team are creating (Shared Lives: Cancer) cannot be classified as a self-help, psychological or educational intervention, it exists primarily as a stand-alone website that the public can browse and interact with at their convenience. Therefore, we needed evidence directly in line with this approach to inform our own work and so consequently these articles were excluded. Overall, 10 studies met the pre-defined eligibility criteria that focused on the development of web-based tools to support PABC.

### Study characteristics

The 10 articles were published between 2012 and 2020 and were undertaken in Australia,[31] Belgium,[32] Vietnam,[33] the UK[34–37] and the USA.[38–40] Five studies focused on people with specific cancer types including survivors of Hodgkin Lymphoma,[38] patients with experience of gynaecological cancers,[34] survivors of oral cancer[39] and patients and survivors of breast cancer,[32 40] while three studies,[31 35 36] included patients with experience of a range of cancer types. Some studies also included family caregivers,[39] intimate partners,[32] healthcare professionals[34 36 38] and researchers[34] alongside people with lived cancer experience. Two of the included studies collected data with carers of people with cancer alongside, academics, charity respresentatives and health professionals.[33 37]

All studies employed an iterative, co-designed methodological approach for the development of web-resources to support PABC. Two of the studies employed a mixed methods research design,[38 40] six used both quantitative and qualitative methods[31 32 35–37 39] and two articles used solely qualitative methods.[33 34] Four studies explored user needs and preferences using focus groups,[32 38] discussion workshops,[33 34] semistructured interviews[33 39] and questionnaires.[32] Three articles explored preferences around the design of the web-based resources using discussion workshops[34 37] and interviews.[31] Seven studies evaluated the usability and/or acceptability of web resources using 'think aloud' cognitive interviews,[36 38–40] focus groups,[35 36] semistructured interviews,[40] structured interviews,[35] acceptability E-scales,[38] readiness scales,[31] website tracking[31 37] and online surveys.[31 40] One study evaluated the feasibility of web-resources using a combination of surveys, questionnaires and structured interviews,[35] and

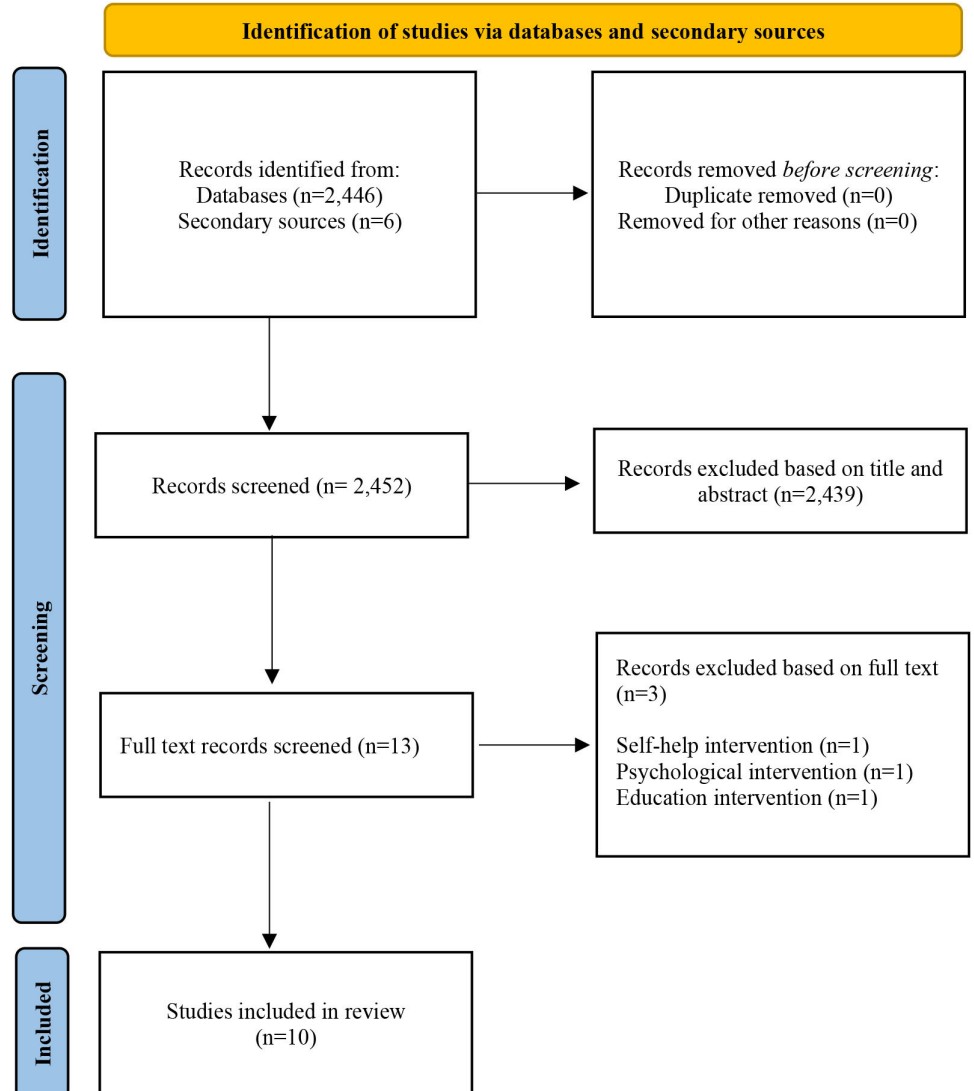

**Figure 1** Study flowchart.[56]

another carried out user testing (separate to usability testing) via interviews and evaluation surveys.[39] See table 1 for further details of the characteristics of included studies.

## Overarching themes

The findings from the ten articles were grouped under the following three areas: (1) exploring current evidence, guidelines and theory, (2) identifying user needs and preferences and (3) evaluating the usability, feasibility and acceptability of resources.

### Exploring current evidence, guidelines and theory

Bradbury *et al*[36] conducted a rapid scoping review to identify the barriers and facilitators to intervention success including the participants needs and attributes and intervention components. Synthesised evidence from the review informed key design objectives including employing an approach that promotes well-being, ensuring the appropriate promotion of behaviour

change, providing easy, timely and tailored information and ensuring an efficient design. These findings were used to establish intervention guiding principles and inform the behavioural analysis and logic model that would underpin resource development. Similarly, Kapoor *et al*[40] conducted a literature review to identify the needs of breast cancer survivors to assist in informing web-resource development. The findings, combined with input from an expert panel, helped to identify core functions to be incorporated into the design of a prototype resource including recording and tracking of quality of life indicators, recording user-reported treatment-related symptoms, viewing breast cancer related medical history, viewing scheduled follow-up visits and generating and displaying customised alerts related to symptoms and quality of life issues. Other studies also reported reviewing patient websites and performing literature reviews but were not explicit on how findings informed web-resource development.[31 37]

**Table 1** Characteristics of included studies

| Study (Country) | Population | Methods/design | Key outcomes |
|---|---|---|---|
| Amweg et al[38] (USA) | Hodgkin Lymphoma Survivors, n=10 Healthcare professionals, n=9 | Mixed methods, user-centred design Two phases: 1. Focus groups 2. Usability testing (cognitive interviews and acceptability E-scale) | *User needs and preferences* (Phase 1) Feedback on participants' specification preferences of website *Usability* (Phase 2) Feedback of preferences and experience of using website Acceptability of website |
| Ashmore et al[34] (UK) | Patients with gynaecological cancer, n=5 Healthcare professionals, n=5 Researchers, n=3 | Qualitative, multidisciplinary co-creation approach Four discussion workshops 1. Establish understanding of available support and treatment 2. Establish key areas for support 3. Website design and requirements 4. Review of initial resource | *User needs and preferences* (Workshops 1 & 2) Establish understanding of available support and treatment Establish key areas of support *Website development* (Workshops 3 & 4) Development of initial resource through creation of a design brief 'wish list' Review of the design of initial resource and identification of recommendations for design team |
| Badr et al[39] (USA) | Oral cancer survivors, n=16 Family caregiver, n=12 | Quantitative and qualitative user-centred design Three phases: 1. Qualitative needs assessment (semistructured interviews) 2. Prototype development 3. Formative evaluation (usability testing—'think aloud' interviews and user testing—interviews and survey) | *Preferences* (Phase 1) Feedback on the unmet needs and preferences for website *Website development* (Phase 2) Development of website prototype *Usability/user testing* (Phase 3) Identify navigational difficulties of website Identify participants' experiences of using website Evaluation survey (attractiveness, controllability, efficiency, intuitiveness, learnability) |
| Bartlett et al[35] (UK) | Patients with cancer, n=259 (Breast, Colorectal, Germ cell, Gynaecology, Haematology, Kidney, Prostate, Sarcoma, Upper gastrointestinal) | Quantitative and qualitative user-centred design Three phases: 1. Website design (focus groups and interviews) 2. Computer and internet survey usage 3. Crossover study (questionnaires and structured interviews) | *Usability* (Phase 1) Patient feedback on initial web resource *Feasibility* (Phase 2) Sociodemographics Computer and internet usage *Usability/feasibility/acceptability* (Phase 2) Web resource activity tracking Usability and acceptability feedback |
| Bradbury et al[36] (UK) | Patients with cancer, n=32 (Patients with breast, colorectal and prostate cancer) Supporters of cancer survivors (nurses, general practitioners, care assistants, cancer charities), n=31 | Quantitative and qualitative evidence, theory and user-centred approach Seven phases: 1. Scoping review 2. Guiding principles 3. Behavioural analysis 4. Logic model 5. Prototype of website 6. Qualitative optimisation study 1 7. Qualitative optimisation study 2 | *Usability/acceptability* (Phases 6 and 7) Feedback of participants experience of exploring website, includes participants' likes, dislikes and recommendations for change |

Continued

**Table 1** Continued

| Study (Country) | Population | Methods/design | Key outcomes |
|---|---|---|---|
| Fennell et al[31] (Australia) | Patients with cancer, n=122 (Bones, breast, cervix, colorectal/bowel, lymphoma, lung, melanoma, ovaries, prostate, testicular, brain) | Quantitative and qualitative user-centred approach Two phases: 1. Website development (interviews) 2. Acceptability testing (website tracking activity, readiness scale, online survey) | *Design* (Phase 1) Feedback on initial website content and design *Usability/acceptability* (Phase 2) Website usage Website acceptability and perceived impact |
| Kapoor et al[40] (USA) | Patient with breast cancer or survivor, n=15 | Mixed methods, evidence, theory and user-centred approach Seven phases: 1. Literature review and expert panel 2. Review of current breast cancer survivorship guidelines and plans 3. Development of decisions 4. Curation of decisions 5. Prototype design and development 6. User feedback (semistructured interviews) 7. Usability testing ('think aloud' and semistructured interviews and online surveys) | *Perceived usefulness* (Phase 6) Identify participants' perception and perceived usefulness of the website *Usability* (Phase 7) Identify the overall usability of the website |
| Pauwels et al[32] (Belgium) | Breast cancer survivors, n=57 Intimate partners, n=28 | Quantitative and qualitative user-centred design Pre and post design (post-questionnaire, website tracking, care needs questionnaire) | *User needs* Assessment of participants' needs for information and support *Design* Evaluation of the content and lay-out of the website. Concepts evaluated include user friendly, well-built, interesting, informative, understandable, new, incomplete, irrelevant, unreliable, too extensive or confusing *Usability* Information gathered about participants' use of the website |
| Santin et al[37] (UK) | Cancer carers, academics, cancer charity representatives, healthcare professionals, n=12 | Quantitative and qualitative co-design approach Two phases: 1. Co-design model – Design of website (workshops and meetings) – Development of prototype 1 – User testing phase 1 (unstructured feedback sessions) – Refining prototype 2. User testing – User testing and refinement (semistructured interviews, web survey, website tracking) – Final development | *Design* (Phase 1) Evaluation and refinement of website design *Usability* (Phases 1 & 2) To gather views and experiences of users' interactions with the web-resource. Evaluate website use through tracking website activity |
| Santin et al[33] (Vietnam) | Informal cancer carers, n=20 Healthcare professionals, n=23 | Qualitative co-design approach Two phases: 1. Identification of needs (interviews and focus groups) 2. Stakeholder verification and refinements (co-design workshops) | *User needs and perspectives* (Phases 1 & 2) Identifying and understanding the needs of informal cancer carers to underpin resource content Learn and agree on shared priorities and resource components between informal cancer carers and healthcare professionals |

In addition to reviewing the available literature, studies also reviewed existing guidelines and theory to inform web-resource development. Kapoor et al[40] conducted a comprehensive review of current breast cancer survivorship guidelines and existing survivorship plans which were used to inform the inclusion of key support information within the web-resource. Badr et al[39] explored the best practices underpinning the management for oral and swallowing complications following radiotherapy, while also reviewing national healthy lifestyle guidelines for cancer survivors and evidence surrounding the self-determination theory. The findings were used to develop a prototype web-resource that specifically focused on promoting survivor and caregiver autonomy, competence and relatedness; by providing tailored information, skill-building education and support services. Other studies also reported reviewing clinical practice guidelines alongside reviewing the academic literature.[31]

## Identifying the needs and preferences for resources

Participants of the included studies emphasised the need for resources that provide comprehensive information on cancer management and survivorship.[32–34 39] The need for clear information on survivorship care with a specific focus on physical, psychosocial, psychosexual and emotional well-being was identified[32 34 38]; in addition to information on adjusting to 'new normal', returning to work, financial management and lifestyle advice.[32 34 39] The inclusion of practical advice and information on the side effects of cancer treatments was viewed as essential[34 39] and participants expressed the need to learn from other survivors and carers through shared experiences and self-care strategies.[33 34 39] Concerns were raised by survivors regarding the risk of secondary cancers and how to communicate with family about experiences of cancer survivorship.[34] The inclusion of a 'Frequently Asked Questions' page was also proposed to ensure a safe space for users to search for specific information.[33 34]

Reported discussions between healthcare professionals focused on the need to ensure resources can be integrated easily into existing digital systems and are accessible across clinical specialities.[38] It was also considered important that participants did not view resources as a substitute for clinical care[38] and that information on family/carer support be included.[34] Caregivers expressed the need for emotional and supportive information on how to cope with cancer in addition to information on cancer side effects and lifestyle advice.[32 33 39] Concerns were also raised regarding the fear of reoccurrence and the need for specific self-care information and better family communication for carers.[39] Caregivers also discussed the inclusion of information regarding cancer causes and treatment, pain management, hospital administration and treatment processes, hospital daily living and signposting to skills training.[33 39]

## Evaluating the usability, feasibility and acceptability of resources

Studies explored the usability, feasibility and acceptability of resources by qualitatively drawing on the users' positive and negative experiences of web-resource interaction. Users viewed web-resources positively and valued their use in providing centralised, easily accessible information to support and facilitate survivorship care.[36–38 40] The content included within web-resources was regarded as useful in managing the consequences of cancer and was viewed as a credible source of information due to its development by trusted experts.[36 37 40] Accessing information through web-resources and video formats was perceived as less burdensome than written information and allowed users to easily access advice.[37] Resource features including providing useful website links, being able to access medical history and tracking quality of life indicators was also perceived as valuable components of web-resources.[40]

While web-resources did provide easy access to information, the content of web-resources was considered impersonal with users expressing the need for more customised and prioritised information[35 37 38 40] that was representative of all genders.[37] Web-resources were found to be too complex with users experiencing difficulties in navigating and understanding the purpose of certain web-features highlighting the importance in developing simple and user-friendly web-resources.[35 38 39] Issues with web-resource design were also experienced with users emphasising the need for more appealing web-designs that use appropriate colour and size of both fonts and paragraphs, include greater cross-links, and incorporate much clearer navigational features.[31 38 39]

Studies also evaluated the usability, feasibility and acceptability of web-resources using a range of quantitative methods. A common approach identified was the use of Likert scale style questionnaires and surveys.[32 39 40] For example, Badr et al[39] reported an overall resource usability score of 80/100 with individual areas rated as attractiveness (4.0/5), controllability (4.2/5), efficiency (4.1/5), intuitiveness (3.9/5) and learnability (3.8/5). Amweg et al[38] employed an acceptability E-scale to objectively identify web-resource acceptability reporting an overall score of 29.8 (a score of <24 was considered an indicator of web-resource acceptability). Other studies also used descriptive questionnaires and surveys with users rating web-resources as easy to use, useful, relevant, necessary and likely to return and recommend.[35 37] Studies were also shown to objectively explore website usability using website analytics.[31 32 35 37] For example, Santin et al[37] reported 2769 unique visits between November 2017 and May 2018 of which 743 were returning visitors. Visitors were shown to access multiple website components including the 'getting through treatment', 'caring for you', 'financial' and 'employment' elements. Peer-led videos were the most frequently accessed content while professional led material, supporting children and the emotional aspects of caring were the least visited.

## DISCUSSION

This rapid review has systematically identified and mapped the peer-reviewed academic evidence that reported on primary data concerning the development of web-based interventions for supporting PABC. Our findings highlight the use of user-centred, co-designed methodological approaches that are underpinned by iterative, but not necessarily sequential, development processes. A common approach used to develop web-based resources involved the initial exploration of the current evidence, guidelines and theory followed by an assessment of user needs and preferences to ensure that web-resources were designed to meet the needs of its users. This was typically proceeded by the evaluation of resources involving usability, feasibility or acceptability testing using a wide range of quantitative, qualitative and mixed methods that often fed back into further resource refinement. While previous reviews focus predominantly on evaluating the effectiveness of web-based resources, this rapid review differs in that it provides important and novel insights into the methodological approaches that underpin the development and implementation of web-based resources to support PABC. Our findings have the potential to assist other researchers who are developing digital resources and will be used by the current research team to inform the development of a web-based support platform (Shared Lives: Cancer)[25 26] that aims to make qualitative research data on lived cancer experiences publicly available via an open access searchable website. Specifically, the findings have made the team aware that the development of digital resources should be informed by the experiences of a wide range of end-users and co-developed where possible and appropriate. The design and content of resources should be centred around the user's needs and preferences and include resource evaluation as part of an iterative approach through usability, feasibility or acceptability testing using a range of different methods. Following the launch of Shared Lives: Cancer, the team will continue to collect data on user experience to ensure its design and content is grounded within the needs of its intended audience.

Ensuring the appropriate design of web-based resources is a critical component of website development[41] in which the use of iterative, co-designed methods is strongly advocated,[42 43] especially with respect to cancer care.[44] This is supported by previous evidence that demonstrated the engagement of stakeholders throughout the developmental process ensures that digital tools are firmly grounded within the user's needs, which consequently improves usability and increases user engagement.[45–47] However, there must be an appreciation that users will have varying levels of digital literacy and this needs to be considered when designing and delivering digital resources. Existing research has shown that poor digital literacy is linked with computer anxiety and barriers to internet use among PABC.[48 49] Therefore, resources should be accessible and lay friendly to encourage engagement with people who have lower levels of digital literacy. At the same time, there will and continues to be PABC who prefer non-digital support for a variety of reasons.

Therefore, it is important that face-to-face support is maintained as digital services continue to be widely rolled out as a consequence of both the COVID-19 pandemic and global healthcare policies.

The findings from this review also emphasise the importance of collecting data on usability, feasibility and acceptability, which are widely considered as important elements when developing web-based resources. An important decision future researchers may face during the ongoing development of digital resources is deciding how these areas will be measured. In line with evidence concerning usability and acceptability testing,[50 51] our findings point towards employing the use of a wide range of quantitative and qualitative methods and where possible should consider a combination of methodologies.[52] While we identify key assessment methods including website analytics, E-scales, questionnaires, 'think aloud' interviews, semistructured interviews, focus groups and workshops, future research should also consider other methods including more objective and automated methods, especially in the context of usability testing.[50 52]

The development and implementation of digital tools has enormous potential in supporting future healthcare services through transforming the way individuals engage with services and professionals, advancing efficient care coordination and allowing individuals to better manage one's health and well-being.[53–55] The use of digital technology is now considered a fundamental element that will underpin many of the proposed changes as part of the NHS long-term plan,[11] including desires to facilitate better care and support for individuals at home through the use of digital health tools. As the NHS looks to transform and adapt over the next decade, it is important to consider digital health technologies as a potential solution to improve and strengthen aspects or cancer care.[44] The findings of the current review provide important methodological insight that should be used to develop emerging digital health technologies that may help transform and support future healthcare services.

A strength of this review is that it allowed for a rapid synthesis of the current evidence needed to provide timely information to inform the decision-making process surrounding the development and implementation of a novel digital support resource (Shared Lives: Cancer)[25 26] as part of an externally funded project. It provides important insight into the methodological approaches used to develop web-based resources which may be used to guide and inform the design of future digital resources. A limitation of the current review was the lack of consistency and uniformity across outcome measurement tools of included studies, making it challenging to compare and interpret findings. While rapid reviews are key in synthesising timely and informative evidence, it is recognised that the accelerated review process is not subject to the same robustness as a full systematic review. The current rapid review used a streamlined review process that restricted literature searches to one database only and omitted the inclusion of assessing risk of bias. We would encourage other researchers who are developing this work further to conduct a full systematic review that also includes a quality assessment

of the academic literature. It is therefore acknowledged that the methodology of the current study is less comprehensive and as a consequence the results may be more susceptible to bias and error.

## CONCLUSION

This research adopted a rapid review approach as there is a timely need for an evidence synthesis to support and inform the development of an ongoing project to design an online web-based platform (Shared Lives: Cancer).[25 26] The findings of this rapid review provide an important insight into the methodological approaches used to underpin the development of web-based interventions to support PABC. The evidence generated from this review has the potential to inform and guide future research endeavours when considering the development and implementation of digital resources.

**Acknowledgements** The authors would like to acknowledge the ongoing support of the East Midlands Cancer Alliance and Macmillan Cancer Support with the development of Shared Lives: Cancer.

**Contributors** SC, DN, HG and RK conceptualised and designed the review. SC reviewed titles, abstracts, full-text articles and extracted data with all data extraction verified by DN. SC and DN prepared the initial manuscript. HG, RK, KM and MG reviewed and edited the final manuscript. SC is responsible for the overall content of this study as guarantor.

**Funding** This research was supported by the National Institute for Health Research (NIHR) Clinical Research Network (CRN) East Midlands Targeted Funding Call (Grant number/Award number: N/A).

**Competing interests** None declared.

**Patient and public involvement** Patients and/or the public were not involved in the design, or conduct, or reporting, or dissemination plans of this research.

**Patient consent for publication** Not applicable.

**Provenance and peer review** Not commissioned; externally peer reviewed.

**Data availability statement** All data relevant to the study are included in the article or uploaded as supplementary information.

**ORCID iD**
Samuel Cooke http://orcid.org/0000-0002-3027-7807

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
