## [Reviewer comments · BMJ Open]

ARTICLE DETAILS

TITLE (PROVISIONAL)	A rapid systematic review on developing web-based interventions to support people affected by cancer.
AUTHORS	Cooke, Samuel; Nelson, David; Green, Heidi; McPeake, Kathie; Gussy, Mark; Kane, R

VERSION 1 – REVIEW

REVIEWER	Katapodi, Maria C. University of Basel, Clinical Research
REVIEW RETURNED	27-Apr-2022

GENERAL COMMENTS	The manuscript presents a rapid review that aims to identify and synthesize studies that present information on the development and pilot evaluation of websites targeting cancer survivors. Overall the manuscript is well-written, information is presented clearly, the introduction presents clearly the significance of digital health in advancing care of people with cancer, and findings are presented clearly. Table 1 is of exceptionally good quality. However, there are some issues that should be further clarified. The authors should define better what they mean by saying that the websites are designed "to support" people affected by cancer. What kind of support are they looking at? The term "support" is very generic and the reader is left wondering what is the target of the study. Along these lines, the three excluded studies in Figure 1 are supportive interventions (i.e., self-help intervention, psychological intervention, and educational intervention). Why were these studies excluded, since they look like they provide support to people affected by cancer. Page 7, Lines 51-53 the authors mention that titles and abstracts were reviewed by one reviewer, however, "identified discrepancies were resolved via discussion." Can they please clarify? Page 8, Lines 39-40, the authors mention that studies not published in English are excluded. This information is unnecessary, since a publication in English is an inclusion criterion
--

REVIEWER	Hommel, Saar Tilburg University, Department of Communication and Cognition
REVIEW RETURNED	09-May-2022

GENERAL COMMENTS	Thank you for the opportunity to read and review your manuscript. It was a pleasure to read this well-written and clearly defined study. The authors explain the process of a rapid review clearly and they have convinced me as a reader why this could be a valid approach. However, my main concern with the study was that the added value (besides making a case for rapid reviews and informing their own research) is not completely clear to me. As the authors claim that "our findings have the potential to inform and guide researchers when considering the development of future digital health resources", I feel that that could be highlighted more. The results to me feel a bit too
--

	much like a summary of the included papers rather than a structural overview of the current evidence. This could also be due to the fact that no specific assessment criteria were used. For me as a reader (and developer of web-based decision aids) I was therefore left with the question what this review adds (besides me not having to read the 10 included papers). Perhaps the authors could address these points in their manuscript more. 1) One improvement could perhaps be to use a specific critical assessment tool that tells readers about the status of the evidence more. How do they regard the qualitative and quantitative methods used? 2) As the authors mention that they use this review for their own web-based intervention, it would be interesting to know what their main take away would be. How will they develop their web-based intervention based on the evidence they found? Is the evidence they found even sufficient to base guidelines on? The authors mention "The focus of this rapid review is on identifying and exploring the literature on the development of web-based support that is delivered to PABC therefore, a quality assessment of included articles was not deemed appropriate.", but critically evaluating literature is a key part of formulating guidelines (how else will you determine what strategies/components to use and what strategies/components to disregard?) relating to the second aim of the manuscript. 3) In the abstract the authors mention how the evidence was grouped (in three themes), but they do not highlight the most important findings within these themes.
--	--

VERSION 1 – AUTHOR RESPONSE

Reviewer #1	The authors should define better what they mean by saying that the websites are designed "to support" people affected by cancer. What kind of support are they looking at? The term "support" is very generic, and the reader is left wondering what is the target of the study.	The following sentence has been added to the second to last paragraph of the introduction to better define what we mean by support: "This will explicitly include resources designed to support the physical, mental, and social consequences of cancer."
	Along these lines, the three excluded studies in Figure 1 are supportive interventions (i.e., self-help intervention, psychological intervention, and educational intervention). Why were these studies excluded, since they look like they provide support to people affected by cancer?	Thank you, you are correct that these do provide support to people affected by cancer, but it should be noted that these were excluded as the resource the team are creating (Shared Lives: Cancer) is not a self-help, psychological, or educational intervention, it exists primarily as a stand-alone website that the public can browse and interact with at their convenience. Therefore, we feel it is justified that these three articles remain excluded from this rapid review. We have added in the following three sentences to the search results paragraph to further highlight and justify our decision: "Three did not meet the pre-specified inclusion criteria as these were self-help, psychological, and educational supportive interventions. The resource the team are creating (Shared Lives: Cancer) cannot be classified as a self-help, psychological, or educational intervention, it exists primarily as a stand-alone website that the public can browse and interact with at their convenience. Therefore, we needed evidence directly in line with this approach to inform our own work and so consequently these articles were excluded"

	Page 7, Lines 51-53 the authors mention that titles and abstracts were reviewed by one reviewer, however, "identified discrepancies were resolved via discussion." Can they please clarify?	Whilst SC did independently screen for title/abstract, discussion between authors did take place where there was uncertainty over the eligibility of an article. We have removed the sentence "Identified discrepancies were resolved via discussion" from the manuscript and instead have included the following sentence to the last paragraph in the search strategy section: "Where there was uncertainty about the inclusion of an article after title and abstract screening the first author (SC) discussed this with the second author (DN) to reach a final decision."
	Page 8, Lines 39-40, the authors mention that studies not published in English are excluded. This information is unnecessary since a publication in English is an inclusion criterion.	The sentence "Any publications that were not written in the English language were excluded from this review" has been removed from the exclusion criteria paragraph.
Reviewer #2	As the authors mention that they use this review for their own web-based intervention, it would be interesting to know what their main take away would be. How will they develop their web-based intervention based on the evidence they found? Is the evidence they found even sufficient to base guidelines on?	Thank you, we have added the following two sentences to the end of the first paragraph of the discussion to show how this has influenced our own work: "Specifically, the findings have made the team aware that the development of digital resources should be informed by the experiences of a wide range of end-users and co-developed where possible and appropriate. The design and content of resources should be centred around the user's needs and preferences and include resource evaluation as part of an iterative approach through usability, feasibility, or acceptability testing using a range of different methods. Following the launch of Shared Lives: Cancer, the team will continue to collect data on user experience to ensure its design and content is grounded within the needs of its intended audience."
	The authors mention "The focus of this rapid review is on identifying and exploring the literature on the development of web-based support that is delivered to PABC therefore, a quality assessment of included articles was not deemed appropriate.", but critically evaluating literature is a key part of formulating guidelines (how else will you determine what strategies/components to use and what strategies/components to disregard?) relating to the second aim of the manuscript. One improvement could perhaps be to use a specific critical assessment tool that tells readers about the status of the evidence more. How do they regard the qualitative and quantitative methods used?	Thank you for this suggestion. Given we used a rapid review approach we worked against a predetermined study protocol that was registered on the Open Science Framework. The protocol was clear that our primary aim was to identify and map the existing evidence in a timely manner to inform the development of our own resource. It was also explicit that a quality assessment would not be performed. The omission of a quality assessment is in line with the methodological approach taken by other rapid systematic reviews that aim to synthesise evidence quickly (see Haby et al, 2016 heavily cited work on this which we have now cited). However, we take on board this valuable point and have now added it as a limitation. We encourage other researchers continuing this work via the methods of a full systematic review to incorporate a full quality assessment. We have added the following sentence at the end of the quality assessment section: "The omission of a quality assessment was in line with the methodological approach taken by other systematic reviews where the focus is on producing evidence quickly (Haby et al. 2016)." We have added the following sentence in the last paragraph of the discussion: "We would encourage other researchers who are developing this work further to conduct a full systematic review that also includes a quality assessment of the academic literature"
	In the abstract the authors mention how the evidence was grouped (in three themes), but they do not highlight the most important findings within these themes.	We have added the following two sentences summarising the most important findings identified within themes: "Resources should be informed by the experiences of a

		wide range of end-users taking into consideration current guidelines and theory early in the design process. Resource design and content should be developed around the user's needs and preferences and evaluated through usability, feasibility, or acceptability testing using quantitative, qualitative, or mixed methods.”
--	--	---

VERSION 2 – REVIEW

REVIEWER	Hommel, Saar Tilburg University, Department of Communication and Cognition
REVIEW RETURNED	29-Jun-2022

GENERAL COMMENTS	Thank you for the opportunity to review your revision. I am happy with the way the authors integrated my comments and have no further remarks. I wish the authors good luck with their interesting project on "Shared lives: Cancer" and look forward to reading more about it in the future.
---